# Gene Co-Expression Analysis of Human *RNASEH2A* Reveals Functional Networks Associated with DNA Replication, DNA Damage Response, and Cell Cycle Regulation

**DOI:** 10.3390/biology10030221

**Published:** 2021-03-13

**Authors:** Stefania Marsili, Ailone Tichon, Deepali Kundnani, Francesca Storici

**Affiliations:** Georgia Institute of Technology, School of Biological Sciences, Atlanta, GA 30332, USA; marsili@gatech.edu (S.M.); eilonti1@gmail.com (A.T.); dkundnani3@gatech.edu (D.K.)

**Keywords:** RNASEH, *RNASEH2A*, co-expression, genotype-tissue expression, cell cycle, cancer

## Abstract

**Simple Summary:**

*RNASEH2A* is the catalytic subunit of the ribonuclease (RNase) H2 ternary complex that plays an important role in maintaining DNA stability in cells. Recent studies have shown that the *RNASEH2A* subunit alone is highly expressed in certain cancer cell types. Via a series of bioinformatics approaches, we found that *RNASEH2A* is highly expressed in human proliferative tissues and many cancers. Our analyses reveal a possible involvement of *RNASEH2A* in cell cycle regulation in addition to its well established role in DNA replication and DNA repair. Our findings underscore that *RNASEH2A* could serve as a biomarker for cancer diagnosis and a therapeutic target.

**Abstract:**

Ribonuclease (RNase) H2 is a key enzyme for the removal of RNA found in DNA-RNA hybrids, playing a fundamental role in biological processes such as DNA replication, telomere maintenance, and DNA damage repair. RNase H2 is a trimer composed of three subunits, *RNASEH2A* being the catalytic subunit. *RNASEH2A* expression levels have been shown to be upregulated in transformed and cancer cells. In this study, we used a bioinformatics approach to identify *RNASEH2A* co-expressed genes in different human tissues to underscore biological processes associated with *RNASEH2A* expression. Our analysis shows functional networks for *RNASEH2A* involvement such as DNA replication and DNA damage response and a novel putative functional network of cell cycle regulation. Further bioinformatics investigation showed increased gene expression in different types of actively cycling cells and tissues, particularly in several cancers, supporting a biological role for *RNASEH2A* but not for the other two subunits of RNase H2 in cell proliferation. Mass spectrometry analysis of *RNASEH2A*-bound proteins identified players functioning in cell cycle regulation. Additional bioinformatic analysis showed that *RNASEH2A* correlates with cancer progression and cell cycle related genes in Cancer Cell Line Encyclopedia (CCLE) and The Cancer Genome Atlas (TCGA) Pan Cancer datasets and supported our mass spectrometry findings.

## 1. Introduction

The development of high-throughput tools to monitor gene expression levels in a specific cell type and tissue has allowed the characterization of gene expression patterns throughout the human body. Such an approach was utilized in several studies to identify molecular signatures of tissues and cells [1,2,3]. Genes that share a molecular pathway in a given tissue or cell should be co-expressed in a spatial and a temporal manner. Therefore, examining genes that show high co-expression correlation in multiple tissues can be used to identify a shared molecular pathway between those genes. The Genotype-Tissue Expression portal (GTEx) [3] has been used as a suitable platform to identify gene co-expression in the human body based on the availability of the full transcriptome in 53 different human tissues. Gene ontology analysis of the co-expressed genes might shed light on genes with uncharacterized functions and help cover additional functions of genes for which their role is partially known.

*RNASEH2A*, together with RNASEH2B and RNASEH2C, composes the holoenzyme ribonuclease (RNase) H2 [4,5]. RNASEH2B and RNASEH2C are used as the scaffold for *RNASEH2A*, which serves as the catalytic subunit of the RNase H2 complex [6]. The function of RNase H2, together with RNASEH1, is to cleave RNA of RNA-DNA hybrids, which can be formed during transcription [7], DNA replication [8], and repair [9]. RNA-DNA hybrids, if not repaired, can harm the genomic DNA by different mechanisms, such as modifying the DNA structure, blocking DNA replication and transcription, causing hyper-recombination and mutation, or chromosome loss [4,10,11,12,13]. Phenotypically, it has been postulated that the RNA-DNA hybrids mimic the infection of nucleic acids of viral origin, thus activating the innate immune response [14]. An example of a pathologic condition is the Aicardi-Goutières Syndrome (AGS), a severe autosomal recessive neurological disorder with symptoms similar to in-utero viral infection [15,16]. Genetic studies have linked AGS to mutations in the genes composing RNase H2, and it has been shown that mutations silencing intracellular RNases affect microRNA turnover, resulting in the severe clinical consequences in the brain that characterize the clinical feature of AGS [17]. However, no evidence was found that RNASEH1 is linked to the AGS pathology [14], highlighting a marked difference between the function of these two enzymes. Another distinctive feature of *RNASEH2A* comes from examining its expression levels in cancer compared to normal tissues. A previous study showed that, in human mesenchymal stem cells transformed by the over-expression of several oncogenes, *RNASEH2A* was among the genes with the highest and the earliest up fold change in their expression [18]. In addition, increased levels of *RNASEH2A* in cancer compared to normal tissues and cells were also reported in cervical cancer [19], prostate cancer [20], colorectal carcinoma [21], and triple negative breast cancer [22]. In support of this finding, a recent study by Luke’s group [23] highlights a differential cell-cycle regulation for the activity and the level of the *RNASEH2A* orthologous gene in yeast *Saccharomyces cerevisiae* (*RNH201*), peaking at S and G2 phases of the cell cycle, supporting the hypothesis that *RNASEH2A* plays a role in cancer progression. This is in contrast with the yeast *RNASEH1* ortholog, *RNH1*, activity and level of which are not altered throughout the cell cycle [23]. With the goal to better understand the biological role of *RNASEH2A* in cancer, we analyzed the prevalence of patients with copy number alterations in the *RNASEH2A* gene in The Cancer Genome Atlas (TCGA) Pan Cancer studies in different cancer tissues, hypothesizing that identifying patterns of its expression can shed new insights on its function.

## 2. Materials and Methods

### 2.1. Co-Expression Correlation Analysis

We obtained expression data from GTEx portal v7 [3] (https://gtexportal.org/home/datasets; accessed date 21 September 2019) which were obtained through RNA-Seq for 53 different tissues. The dataset contained total 56,202 transcripts, out of which 42,548 transcripts had greater than zero transcripts per million (TPM) values in at least one tissue of the 53 different tissues. We calculated Pearson correlation coefficients for every transcript with *RNASEH* transcripts data using R [24]. Then, we analyzed the gene ontology (GO) term of the top 2% or 5% co-expressed genes using Gene Ontology enRichment anaLysis and visuaLizAtion tool (GOrilla) [25].

### 2.2. Co-Expression Correlation Analysis Verification with STRING Database

To validate the correlation co-expression coefficients of different genes with *RNASEH2A* in GTEx tissues, we used protein co-expression data obtained from STRING protein–protein association network v11 [26]. We looked at top correlated genes including *RNASEH2A* in GTEx dataset, which also had at least 20 exclusively co-expressed proteins in STRING protein association network. We used the default filtering criteria of medium confidence score in STRING database, selecting only co-expressed proteins. With 10 such transcripts (including *RNASEH2A*) and 20 co-expressed proteins for each from the STRING database, we ended up with a list of 200 proteins. We removed any duplicate proteins listed and counted how many proteins were in the top five percentile of the Pearson’s correlation coefficient distribution from the co-expression correlation analysis that was to be validated.

### 2.3. Gene Expression Analysis in Cancer

Data were obtained from Cancer RNA-Seq Nexus (CRN) database providing phenotype-specific coding-transcript/lncRNA expression profiles, and mRNA-lncRNA co-expression networks in cancer cells [27] were used to examine the expression level of *RNASEH* genes in different type of cancers compared to normal tissues and in cancers at different stage of progression.

### 2.4. RNASEH2A Copy Number Alterations Analysis in TCGA Pan Cancer Dataset

To analyze copy number alterations (CNAs) in *RNASEH2A*, we used The Cancer Genome Atlas (TCGA) Pan Cancer studies [28] involving 32 studies and 10,967 patients through cBioPortal [29,30]. After filtering samples containing both copy number alterations generated by Genomic Identification of Significant Targets in Cancer (GISTIC) and RNA-Seq data, processing, and normalizing by RNA-Seq by expectation maximization (RSEM), data for 9889 patients were used for final analysis.

### 2.5. Cell and Protein Extracts

Human embryonic kidney, HEK293 were grown in Dulbecco’s Modified Eagle Medium (DMEM) medium (Corning Inc., Corning, NY, USA, cat # 45000-304) containing 10% fetal bovine serum (FBS) (Sigma-Aldrich, St. Louis, MO, USA, cat # F0926) and 1% penicillin-streptomycin (Fisher Scientific, Hampton, NH, USA, cat # 15-140-122) in 5% CO_2_ incubator at 37 °C and were passaged once a week to maintain cell growth. Whole cell lysate from HEK293 was extracted with NP-40 lysis buffer (50 mM Tris-HCL pH 7.4, 150 mM NaCl, 0.5 mM EDTA, 0.5% NP-40) together with a cocktail of protease inhibitor (Complete EDTA-free, Roche Applied Science, Indianapolis, IN, USA) followed by centrifugation at 14,000 rpm for 15 min at +4 °C to pellet the debris.

### 2.6. Plasmids and Transfection

To over-express *RNASEH2A* in HEK293 cell line, we used the *pEGFP-RNASEH2A* plasmid (Addgene, Watertown, MA, USA, cat # 108700). As control, we cloned the *eGFP* plasmid by restricting the *pEGFP-RNASEH2A* using HincII (New England Biolabs, Radnor, PA, USA, cat # R0103S) that cut upstream and downstream of the *RNASEH2A* gene but left the *EGFP* intact. Then, the plasmid was ligated by T4 ligase (New England Biolabs, Ipswich, MA, USA, cat # 101228-176) and transformed into Agilent XL1-Blue Electroporation-Competent Cells (Agilent, Santa Clare, CA, USA, cat # 50-125-045) by electroporation using Gene Pulser Xcell electroporation system (Bio-Rad, Hercules, CA, USA, cat # 1652660). Isolated colonies were grown in Luria-Bertani (LB) medium overnight; plasmids were extracted using the miniprep kit (Fisher Scientific, Hampton, NH, USA, cat # FERK0503) and sent to DNA sequencing to verify the plasmid sequence.

For transient transfection, HEK293 cells were plated at 1–3 × 10^6^ per 10 cm plate. The following day, 5 µg of the plasmids were incubated with 25 µL of the Lipofectamine transfection reagent (Fisher Scientific, Hampton, NH, USA, cat # 11668019) for 10–15 min at room temperature in Opti-MEM media (Fisher Scientific, Hampton, NH, USA, cat # 31985062), and the plasmid-reagent mix was distributed to the cells. The next day, transfected cells were fed with fresh media and harvested 48 h after transfection.

### 2.7. Co-Immunoprecipitation

Whole cell lysate (1 mg) from HEK293 cells transfected with either the *eGFP* or the *pEGFP-RNASEH2A* plasmid was incubated with 30 µL of GFP-Trap Magnetic Agarose beads (ChromoTek, Planegg-Martinsried, Germany, gtma-10 lot # 80912001MA) at 4 °C with end to end rotation for 1 h. Then, the magnetic beads–GFP protein complex were separated from the rest of the protein extract using a magnetic stand. The beads were washed twice with wash buffer I (50 mM Tris-HCL pH 7.5, 150 mM NaCl, 0.25% NP-40) and an additional 2 washes with wash buffer II (50 mM Tris-HCL pH 7.5, 150 mM NaCl). Then, the beads were resuspended in SDS loading buffer and boiled at 95 °C for 10 min. Finally, the beads were centrifuged at 13,000 rpm for 5 min at room temperature, and the eluted proteins were collected and used for either Western blot or mass spectrometry analysis.

### 2.8. Protein Analysis (Western Blot)

Total protein concentrations from HEK293 lysate were quantified (Bradford Protein Assay, Pierce Biotechnology, Rockford, IL, USA), and 15 µg of the protein extracts was separated on a 10% SDS-polyacrylamide electrophoresis gel, transferred to 0.45 µm nitrocellulose membrane (Amersham, Buckinghamshire, UK, cat # 10120-006). The blotted membrane was then blocked in 5% non-fat dry milk tris-buffered saline and Tween 20 (TBS-T) (10 mM Tris-Cl (pH 7.5), 100 mM NaCl, 0.1% Tween-20) at room temperature for 1 h and incubated overnight with specific primary antibodies: *anti-RNASEH2A* (Santa Cruz Biotechnology, Dallas, TX, USA, 1:1000) and anti-GFP (B2) (Santa Cruz Biotechnology, Dallas, TX, USA, 1:1000). After washing 3 × 10 min with TBS-T, the membrane was incubated with a mouse secondary antibody conjugated with horseradish peroxidase (Thermo Fisher Scientific, Waltham, MA, 1:10,000) at room temperature for 1 h. Following washing 3 × 10 min with TBS-T, protein signals were visualized using the electrochemiluminescence (ECL) method according to the manufacture’s recommendations (Pierce™ ECL Western Blotting Substrate, Thermo Fisher Scientific, Waltham, MA, USA) and exposed on autoradiograph films (Denville Scientific, South Plainfield, NJ, USA).

### 2.9. Mass Spectrometry Analysis

The gel lanes were excised from the gel, and the proteins were reduced, alkylated, and digested with trypsin as previously described [31]. The peptides were analyzed by nano-LC-MS/MS and peptide identification as previously described [32]. The raw files were searched using the Mascot algorithm (v2.5.1) against a protein database constructed by combining the bovine reference database (Uniprot.com, 37,941 entries, downloaded 22 May 2019), the human reference database (Uniprot.com, 73,911 entries, downloaded 22 May 2019), the *GFP-RNASEH2A* protein, and a contaminant database (cRAP, downloaded 21 November 2016 from http://www.thegpm.org) via Proteome Discoverer 2.1. A 1% false discovery rate (FDR) (“High Confidence”) was used for both the peptides and the proteins. At least 3 peptide sequences for the *RNASEH2A* protein were considered for the analysis.

Spectral count data were analyzed for mass spectrometry runs on two sets of immuno-precipitation experiments described previously in the methods. Counts in each run were normalized based on mean of total spectral counts in all runs in a respective set of the experiment. We then filtered hits specific to *Homo sapiens,* added a count of 1 and log2 transformed the spectral counts. We then used these transformed data to get differentially present protein candidates in immunoprecipitation product of HEK293 with *GFP-RNASEH2A* vs. HEK293 with GFP using linear fitting and empirical Bayes method from limma, a bioconductor package [33,34,35]. To get the final list of protein candidates, we used the filtering criteria of log fold change > 1.5 and FDR adjusted *p* value < 0.05.

### 2.10. RNASEH2A Correlation with Cancer Progression and Cell Cycle Related Genes in CCLE and TCGA Pan Cancer Dataset

To validate the results of our study, we further made use of 2 large cancer related datasets to investigate the correlations between *RNASEH2A* and 39 other genes, including common cancer proliferative markers with few genes involved in each cell cycle phase as well as genes upregulated and downregulated in cancer as controls. We also included the genes for the proteins we identified as putative binding partners from mass spectrometry analysis. We used Broad Institute Cancer Cell Line Encyclopedia (CCLE) dataset containing RNA-Seq expression levels in Reads Per Kilobase of transcript, per Million mapped reads (RPKM) in 1019 cancer cell lines from 26 different tissues of origin [36]. The second dataset we used was the TCGA Pan Cancer Study containing batch normalized RNA-Seq expression levels quantified by RNA-Seq by Expectation Maximization (RSEM) in 32 studies from 10,071 cancer patients. [29]. We added a count of 1 to the values and log transformed the expression counts in both the datasets. These transformed data were used to draw correlations between gene expressions with rcorr function from Hmisc package in R [37]. We also used hierarchical clustering the group the genes in the correlation plot.

## 3. Results

### 3.1. RNASEH2A Is Co-Expressed with Genes that Function in Cell Cycle Regulation

To study *RNASEH2A* function based on co-expressed genes, we used an approach (Figure 1) that utilizes data obtained from GTEx database [3] consisting of RNA-seq output of 42,548 nuclear and mitochondrial transcripts including coding, non-coding, and isoforms (raw data in Appendix A).

After analysis of the top 2% genes (851) with highest correlation coefficient, using the GOrilla analysis tool [25], we observed a significant enrichment of genes involved in the biological processes of cellular response to DNA damage stimulus (*p* = 2.77 × 10^−5^) and DNA replication (*p* = 2.66 × 10^−4^), compatible with the known functions of *RNASEH2A.* In addition, we found enrichment of genes involved in mitotic cell-cycle process (*p* = 2.80 × 10^−14^), regulation of microtubule cytoskeleton organization (*p* = 1.39 × 10^−7^), and chromosome segregation (*p* = 9.00 × 10^−4^) (Table 1 and full data in Appendix A). For molecular function, we found enrichment of catalytic activity on DNA (*p* = 9.59 × 10^−6^) but also microtubule binding (*p* = 4.25 × 10^−5^) and kinase binding (*p* = 3.28 × 10^−4^) (Table 1 and full data in Appendix A). For cellular component, we found gene enrichment in chromosomal part (4.06 × 10^−8^), spindle pole (*p* = 1.13 × 10^−6^), as well as nucleoplasm (*p* = 3.52 × 10^−6^), midbody (*p* = 5.01 × 10^−6^), microtubule organizing center (*p* = 5.13 × 10^−6^), and condensed chromosomes outer kinetochore (*p* = 8.72 × 10^−6^) (Table 1 and full data in Appendix A). Based on these results, we suggest that *RNASEH2A* is involved, among others, in three main functional networks related to DNA replication, DNA damage repair, and regulation of chromosome segregation in mitosis.

To validate the correlation of transcripts in GTEx tissues with *RNASEH2A* at the protein expression level using the STRING protein–protein association network, we looked at 10 transcripts having high correlation with *RNASEH2A* (including *RNASEH2A*) that also have at least 20 known and characterized co-expressed proteins in the STRING network (full list of proteins in Appendix A). Out of the list of 200 proteins (Appendix A), 97 were unique (Appendix A), out of which 87 have Pearson’s correlation coefficient, R > 0.696 falling in the top five percentile (Appendix A), and 68 have Pearson’s correlation coefficient, R > 0.842 falling in the top one percentile (Appendix A) of correlation coefficient distribution previously represented in Figure 1.

Analysis of the top 2% co-expressed correlated genes with *RNASEH1* revealed functional network of genes that are involved in RNA binding (Appendix A for functional network analysis and Appendix A for complete list of *RNASEH1* co-expressed genes). After filtering top 5% correlated genes, we could identify processes such as regulation of telomerase RNA and telomeres maintenance (Appendix A), which were previously reported [38,39]. When we analyzed the top 2% co-expressed correlated genes with *RNASEH2B*, we found the functional network of genes involved in DNA replication (Appendix A for functional network analysis and Appendix A for complete list of *RNASEH2B* co-expressed genes), as expected for this RNase H2 subunit. For *RNASEH2C,* we could not find any functional network either in the 2% or the 5% analyses, as there were no GO terms with an enrichment p-value below the specified value *p* < 0.001 (data not shown).

### 3.2. RNASEH2A Expression Is Increased in Actively Cycling Cells and Tissues

Next, we studied the expression levels of *RNASEH* genes using the GTEx v.7 database, which allowed us to compare the expression levels of *RNASEH1*, *RNASEH2A*, *RNASEH2B,* and *RNASEH2C* genes in 53 different human tissues, including two cell lines (Table 2). We noticed that tissues that are part of the reproduction system (such as testis, uterus, and cervix) as well as transformed lymphocytes and transformed fibroblasts tend to have a higher expression level of *RNASEH* genes, while low levels were found mainly in tissues with low proliferation and cell turnover capacity, such as most regions of the brain, the kidney, and the heart (Table 2).

We then examined the expression level of the *RNASEH* genes in cancer using data obtained from Cancer RNA-seq Nexus (CRN) [27] in 29 different randomly selected tumors in different tissues at different stages of cancer progression compared to their healthy controls (complete list of tumors in Appendix A and raw data in Appendix A). To validate our approach, we also examined the expression level of *MYBL2* and *SCARA5* genes, which are reported to be upregulated and downregulated in cancer, respectively [40]. As expected, *MYBL2* was mostly upregulated in cancers such as triple negative breast cancer, bladder urothelial carcinoma stages 2 and 4, and lung adenocarcinoma stages 2 and 4, while *SCARA5* was downregulated in cancers such as bladder urothelial carcinoma stages 2 and 4; thyroid carcinoma stage 1, and rectum adenocarcinoma stage 2A (Figure 2). *RNASEH2A* was the only *RNASEH* gene that showed increased expression levels in all 29 tumors relative to the control tissues (Figure 2).

To study the levels of *RNASEH* genes throughout the progression of cancer, we examined the expression levels of the genes at different stages of lung squamous adenocarcinoma, breast invasive carcinoma, and bladder urothelial carcinoma. In all these cancers, the expression of *RNASEH2A* changed drastically from a normal stage to early cancer stages and remained high with the progression of the disease (Figure 3 and Appendix A and raw data in Appendix A). No other gene beside *MYBL2* was upregulated at the different stages studied. These results imply that *RNASEH2A* might have a role in cancer etiology or transformation from normal tissue to cancer tissue, that is dependent on its expression level.

### 3.3. RNASEH2A Gene Amplifications Has Higher Prevalence in Multiple Cancer Types

The analysis of *RNASEH2A* copy number alterations and mRNA expression in various types of cancer (prevalence of *RNASEH2A* CNAs in Table 3, expression summary for CNA types and raw data in Appendix A) revealed that the average expression of *RNASEH2A* is higher in patients with amplifications of the *RNASEH2A* gene and lower in patients with deep deletions when compared to normal diploid samples. This correlates with previous findings in mRNA versus CNA type expression levels in the TCGA Pan Cancer dataset [41]. We also observed that, in 16 out of 35 cancer types/subtypes, percentage of patients with amplifications was higher than percentage of patients with deep deletions, with 10% cancer types having >1% amplifications as copy number alteration. The maximum percentage of patients having *RNASEH2A* amplifications in a given type of cancer was 8.14% for ovarian epithelial tumor, followed by 3.29% in endometrial and 2.63% in adrenocortical carcinoma (Table 3 and raw data in Appendix A).

### 3.4. Mass Spectrometry Analysis Identified RNASEH2A Binding Partners Involved in Mitosis Regulation

To identify proteins that interact with *RNASEH2A*, we over-expressed the control eGFP (Figure 4a, lanes 1–3) and the *eGFP-RNASEH2A* (Figure 4a, lanes 4–6) proteins in HEK293 cells. We performed co-immunoprecipitation (Co-IP) using an anti-GFP antibody (Figure 4a). We also confirmed that both *RNASEH2A* (Figure 4b, lanes 1–2) and GFP (Figure 4b, lanes 3–4) antibodies identify the same *GFP-RNASEH2A* protein (Figure 4b). After confirming the successful pulldown of *RNASEH2A* protein, we analyzed the eluted proteins that were interacting with *RNASEH2A* via mass spectrometry. The analysis revealed a short list of five human proteins with at least three peptide spectrum matches (PSMs) interacting with *RNASEH2A* (Table 4 and full data in Appendix A). RNASEH2B and RNASEH2C were interacting with *RNASEH2A* confirming the validity of the Co-IP experiment. Interestingly, among the proteins interacting with *RNASEH2A*, we identified the T-complex protein 1 subunits theta Chaperonin Containing TCP1 Subunit 8 (CCT8) and DnaJ homologue subfamily A member 1 (DNAJA1) and apoptosis-inducing factor mitochondrion-associated 1 (AIFM1) proteins with a gene correlation coefficient with *RNASEH2A* of 0.84, 0.585, and 0.378 respectively. These three potential RNASH2A binding partners have shown to play a role in cell cycle regulation and mitosis [42,43,44,45].

### 3.5. RNASEH2A Expression Positively Correlates with Cancer Proliferation Markers and Cell Cycle Genes

To further explore the involvement of *RNASEH2A* in cancer, we examined the correlations between *RNASEH2A* and 39 additional genes in CCLE and TGCA Pan Cancer datasets (Table 5 and Appendix A). The correlation analysis uncovered a positive correlation of *RNASEH2A* with genes up-regulated in cancer and a negative correlation with genes that are instead down-regulated. In addition, *RNASEH2A* showed a higher correlation with cluster of genes known to play a role in cancer proliferation as compared to *RNASEH2B*, *RNASE2C*, and *RNASEH1*. We also observed a similar trend for the three putative binding partners of *RNASEH2A* that we identified by mass spectrometry, showing a correlation comparable with that found in the GTEx dataset. The analysis of cell cycle related genes showed a high correlation between *RNASEH2A* and genes involved in Gap 1/Synthesis (G1/S), Gap 2 (G2), G2/Mitosis (M), M, and M/G1 phases and a comparatively lower or no correlation with genes unique to G1 and S phases (Figure 5 and raw data in Appendix A).

## 4. Discussion

Co-expression correlation analysis of genes is a simple approach to suggest functional network of genes. We anticipate that this approach is more accurate for analyzing genes sharing a higher correlation value. In addition, we assume that the genes analyzed should have a functional network involved in a process that is dependent on a spatio-temporal gene expression profile. Cell cycle regulation is a good example for such a process. In fact, the analysis of *CDK1*, a regulator of cyclin B implicated in cell cycle control, shows enrichment of genes involved in mitotic cell cycle regulation, microtubule cytoskeleton organization, and regulation of G1/S transition of mitotic cell cycle (Appendix A and complete list of *CDK1* co-expressed genes in Appendix A). This approach can be applied to other processes such as, for example, RNA regulation in stress granules, which is a well-orchestrated process dependent on time and stress conditions. In this context, *G3BP1*, a RNA helicase that is one of the key assemblers of the stress granules, shows enrichment of co-expressed genes involved in molecular function of RNA binding and RNA helicase activity in a ribonucleoprotein cellular component (Appendix A and complete list of *G3BP1* co-expressed genes in Appendix A).

Among the three *RNASEH* genes, we found that only *RNASEH2A* was involved in the functional network of mitotic cell cycle regulation. One possibility is that *RNASEH2A* has a function independent of RNASEH2B and RNASEH2C, as its distribution in the cell was shown not to be limited to the interaction with RNASEH2B and RNASEH2C [6]. This notion was also suggested by others showing that knocking out *RNASEH2A* in cervical cancer HeLa cells results in an increased sensitivity to ataxia telengiectasia and Rad3-related (ATR) inhibitors compared to knocking out *RNASEH2B* [48]. We propose that *RNASEH2A* levels constitute an additional layer of regulation of *RNASEH2A* activity, which is not dictated by *RNASEH2B* or *RNASEH2C* levels, and this is why the co-expression correlation analysis did not show a functional network of mitotic cell regulation for *RNASEH2B* and *RNASEH2C* genes. Following this notion, it would be intriguing to examine the levels of the three RNase H2 genes throughout the cell cycle in human cells and study how perturbing their levels affects RNase H2 function and/or cell cycle progression. Similar studies have been performed on yeast showing a cell cycle dependent expression of *RNH201* (orthologous gene of *RNASEH2A*), peaking at S and G2 phases [23,49], and being recruited to chromatin/telomeres at the G2 phase [49].

Rejins et al. [50], using an anti-mouse antibody for all proteins of the RNase H2 complex, demonstrated an increase in the expression level of RNase H2 in the mouse blastocyst in all three embryonic layers during gastrulation and showed, in newborns and adults, that the expression becomes restricted to highly proliferative tissues such as intestinal crypt epithelium and testes [50]. Moreover, they reported that RNase H2 levels correlate with the proliferation marker Ki67 [50]. These data are consistent with our findings showing an increase in *RNASEH2A* level in actively cycling tissues as well as different cancer tissues compared to normal ones.

Using our co-expression correlation approach, we identified three main functional networks in which *RNASEH2A* is involved: DNA replication, DNA damage repair, and regulation of chromosome segregation in mitosis. Hyper-proliferation of cells results in excessive addition of RNA Okazaki fragments into the genome during DNA replication, increases the number of events when replicative DNA polymerases insert ribonucleotides instead of deoxyribonucleotides into the genomic DNA [51,52], and increases the rate of chromosomes reduction from 4N to 2N by symmetric segregation into two new daughter cells. All these events correspond to the functional networks identified in this study for *RNASEH2A* and link *RNASEH2A* activity to cell cycle regulation. In support of this notion, we identified several cellular pathways among the highest *RNASEH2A* co-expressed genes such as the minichromosome maintenance (MCM) complex, a replicative eukaryotic helicase that, in yeast, has been demonstrated to be activated upon accumulation of RNA-DNA hybrids at the G2-M checkpoint. This finding suggests that targeting the *RNASEH2A* level and/or activity could prevent the DNA damage occurring during replication, which leads to mitotic catastrophe and cell death [53,54]. The other identified complexes, such as the *NDC80* complex, the *CENP* family, and the *KIF* family, have all been shown to function in cell cycle regulation through regulation of the microtubule filaments with the kinetochore [55,56,57].

Our analyses of expression levels focused on the *RNASEH* genes (*RNASEH1*, *RNASEH2A*, *RNASEH2B,* and *RNASEH2C*) using the GTEx v.7 database and revealed a marked contrast in *RNASEH2A* expression levels between low and high proliferative tissues. *RNASEH2A* showed low expression levels in low proliferative tissues such as brain, kidney (cortex), and heart and high expression levels in highly proliferative tissues such as skin, esophagus, small intestine, testis, cervix, as well as in transformed fibroblasts and transformed lymphocytes. Moreover, in line with the initial study by Flanagan et al. [18], showing elevated expression levels of *RNASEH2A* in transformed mesenchymal stem cells, our analysis of 29 different randomly selected tumors in different tissues compared to their healthy controls revealed that *RNASEH2A* is the only *RNASEH* gene displaying increased expression levels. Further analyses of *RNASEH* expression levels throughout different stages of lung squamous adenocarcinoma, breast invasive carcinoma, and bladder urothelial carcinoma again highlighted specific upregulation of *RNASEH2A* from normal tissues to early stages and more advanced stages of these cancers. It is interesting to note that Dai et al. reported on a possible role of *RNASEH2A* in glioma cell proliferation. Their findings suggested a role of *RNASEH2A* upregulation in cell growth and apoptosis, contributing to glioma-genesis and cancer progression [58]. More recently, *RNASEH2A* overexpression was associated with cancer cell resistance to chemotherapy in vitro and with aggressiveness and poor outcomes in breast cancers of estrogen receptor (ER)-positive subtypes [59]. The analysis of copy number alterations of the *RNASEH2A* gene that we performed in our study further supports the role of *RNASEH2A* in cancer development. We demonstrated that *RNASEH2A* gene amplifications have higher incidence in multiple cancer types when compared to deep deletions and normal diploid samples suggesting *RNASEH2A* as a target for cancer diagnosis and therapy. Interestingly, cancers from tissues with high cell turnover, such as the reproductive system, showed a maximum percentage of patients with *RNASEH2A* amplification, suggesting a possible role of adult stem cells in the overexpression of *RNASEH2A.* In fact, deregulation of the adult stem cell niche is considered a key event in the etiology of cancer [60].

Our mass spectrometry analysis shows five protein candidates that specifically interact with *RNASEH2A*. The top two proteins observed were RNASEH2B and RNASEH2C, confirming the authenticity of the results. Additional proteins that we identified in the context of *RNASEH2A* biology and mitosis regulation were CCT8, DNAJA1, and AIFM1. CCT8 is part of the CCT/TRiC complex that was reported to regulate telomerase function by mediating its trafficking from the cytoplasm to the telomeres [36]. In addition, it was shown to promote chromosome segregation by disassembling checkpoint complex from sister chromatids in the initial phase of mitosis [37], supporting the *RNASEH2A* functional network in mitosis regulation. DNAJA1 belongs to the family of DnaJ heat shock protein family (Hsp40); it has been shown to be activated by E2F transcription factor 1 (E2F1) and to promote cell cycle by stabilizing cell division protein 45 (CDC45) [38]. Of interest, the induction of *RNASEH2A* by E2F1 has been also reported in human papillomavirus cervical cancers [19]. AIFM1 is a well know factor that plays a key role in the apoptosis signaling pathways. A recent study has uncovered a new role of AIFM1 on the proliferation of hepatoma cells and cell cycle arrest [39].

The high correlation of *RNASEH2A* expression with genes that play a role in cancer development and cell cycle progression validate our study, showing that *RNASEH2A* plays a role in cancer transformation and progression. In addition, our final analysis supports the mass spectrometry findings and indicates CCT8, DNAJA1, and AIFM1 as binding partners of *RNASEH2A*.

The analysis and the experimental approaches used in this works have some limitations. In this respect, we made use of the Cancer RNA-Seq Nexus database for which patient information is not available. Factors such as age, gender, and medical treatment versus surgery could have influenced the results of our in-silico analysis. In addition, although mass spectrometry is a highly specific and sensitive technique, further investigation is needed to confirm the specific interactions between the *RNASEH2A* subunit and the candidate proteins found in our experiments. In our in vitro experiments, we used the HEK293 cell line. Depending on a number of factors, this cell line in culture can either show the phenotype of a normal or of a cancer cell line [61]. Although we were able to support our in vitro finding by in-silico analysis, the characterization of the biological role of CCT8, DNAJA1, and AIFM1 in concert with *RNASEH2A* requires a more appropriate model system.

## 5. Conclusions

Overall, our study suggests an emerging role of *RNASEH2A* in cell cycle regulation that appears independent from its function as part of the RNase H2 whole enzyme. The presented findings stimulate new research directions such as investigating the differential expression of *RNASEH2A* during cell cycle phases to better understanding and characterizing the function of *RNASEH2A* in cell proliferation in healthy, adults stem cells and cancer cells and to support further exploring *RNASEH2A* as a target for cancer diagnosis and therapy.

## Figures and Tables

**Figure 1 biology-10-00221-f001:**
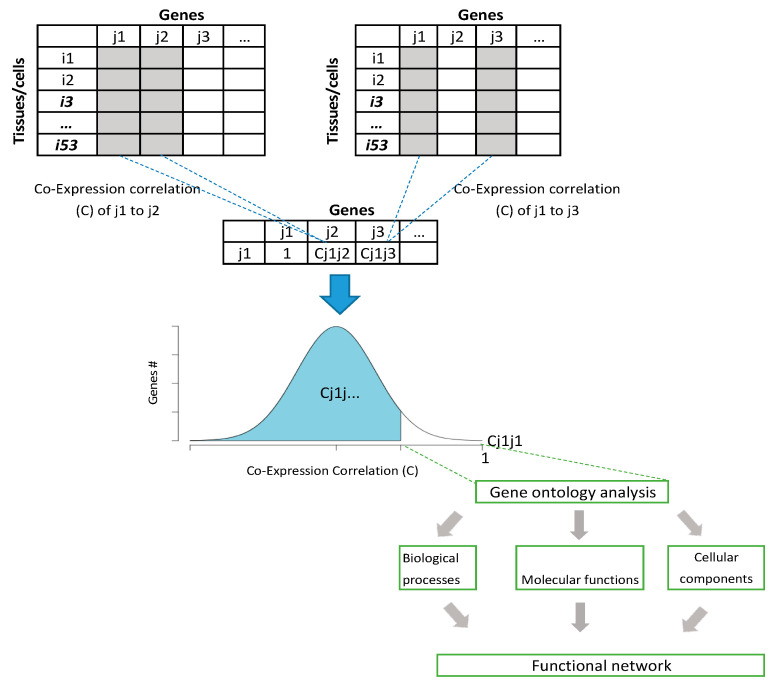
Co-expression correlation analysis approach. Diagram showing the process of analyzing gene functional networks using co-expression analysis. The analysis is based on any database that shows expression levels on multiple tissues and cells from the same organism.

**Figure 2 biology-10-00221-f002:**
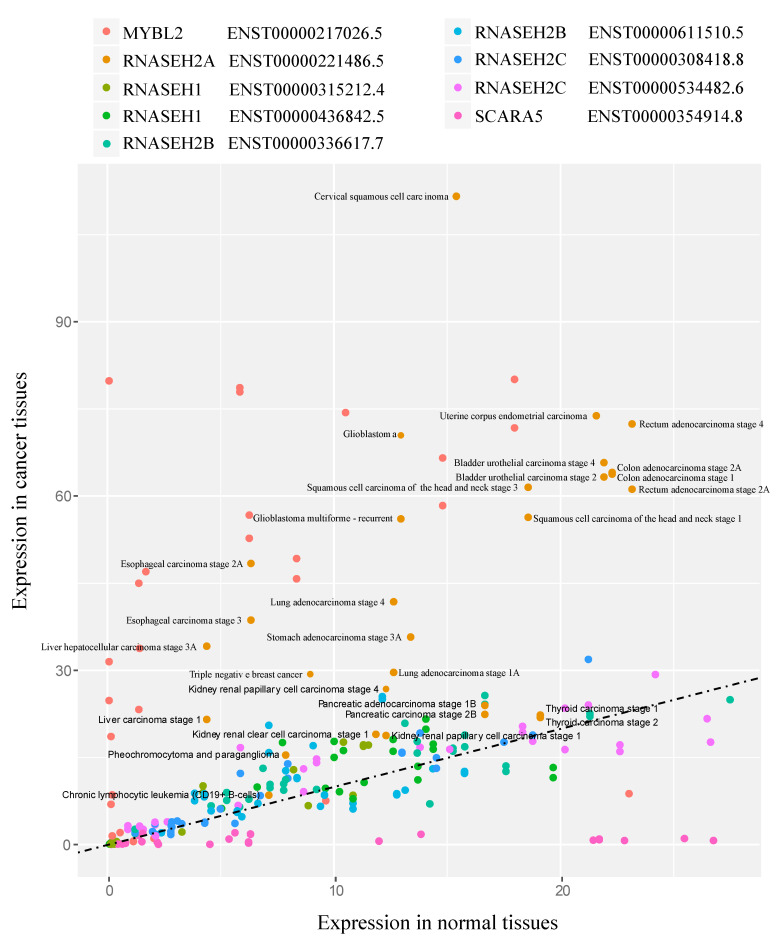
*RNASEH2A* gene expression in cancer tissues. Expression levels in TPM and only for triple negative breast cancer and chronic lymphocytic leukemia in fragments per kilobase million (FPKM) in 29 different cancer tissues compared to non-cancerous tissue controls. Data obtained from Cancer RNA-seq Nexus (CRN). Regression line represents equal expression levels in cancer and normal tissues.

**Figure 3 biology-10-00221-f003:**
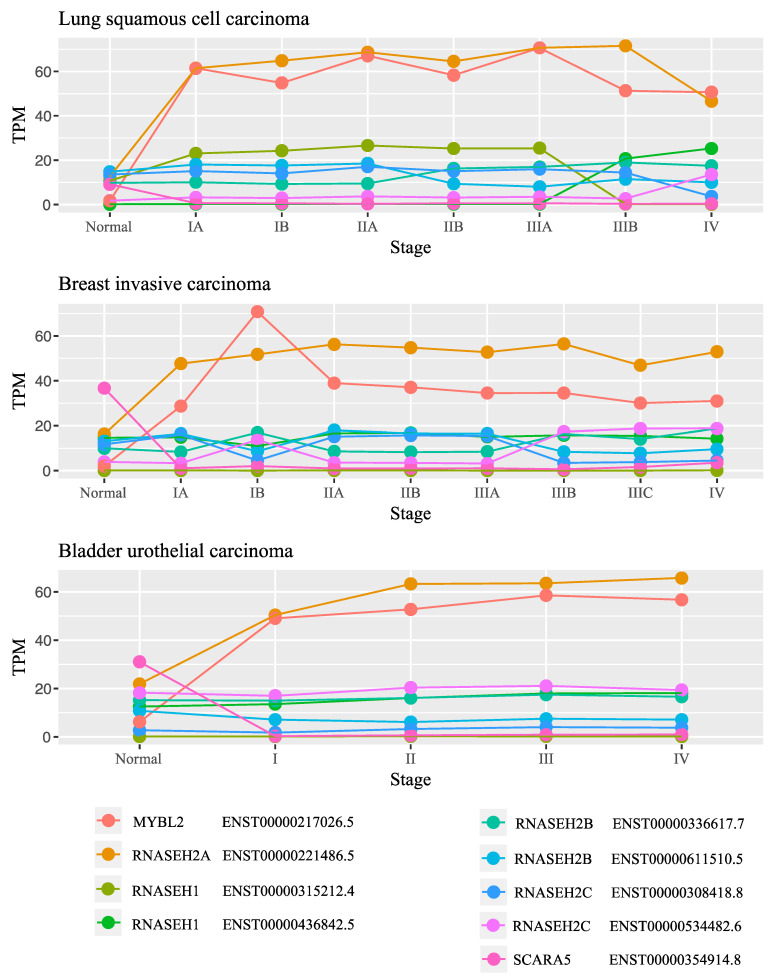
*RNASEH2A* levels in normal and at different stages of lung squamous cell carcinoma, breast invasive carcinoma, bladder urothelial carcinoma (top to bottom, respectively). Expression levels of different transcripts of *RNASEH* genes, including *MYBL2* and *SCARA5*, were examined in in TPM.

**Figure 4 biology-10-00221-f004:**
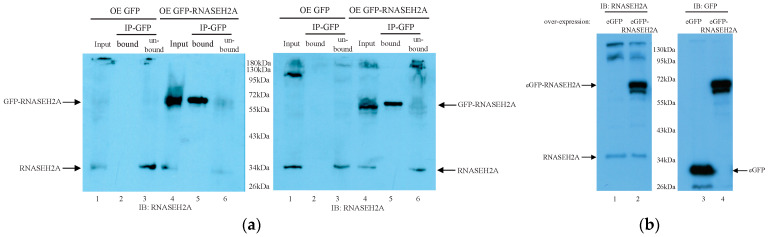
Co-immunoprecipitation and antibodies verification: (**a**) co-immunoprecipitation of *RNASEH2A*. Western Blot of the GFP co-immunoprecipitation. The two panels indicate two independent experiments. HEK293 cells were overexpressed (OE) with the *eGFP* vector (lanes 1–3 in each panel) or with the *eGFP-RNASEH2A* vector (lanes 4–6 in each panel). IP: anti-GFP, IB: *anti-RNASEH2A*. Arrows indicate endogenous *RNASEH2A* (lower arrow) and exogenous *eGFP-RNASEH2A* (upper row); (**b**) *RNASEH2A* and GFP antibodies verification. Western Blot of the *anti-RNASEH2A* (left panel) and the anti-GFP (right panel). In the two panels, HEK293 cells were OE with *eGFP* vector (lanes 1, 3) or *eGFP-RNASEH2A* vector (lanes 2, 4). In the left panel, arrows indicate endogenous *RNASEH2A* (lower arrow) and exogenous *eGFP-RNASEH2A* (upper row). In the right panel, arrows indicate eGFP (lower arrow) and *eGFP-RNASEH2A* (upper row).

**Figure 5 biology-10-00221-f005:**
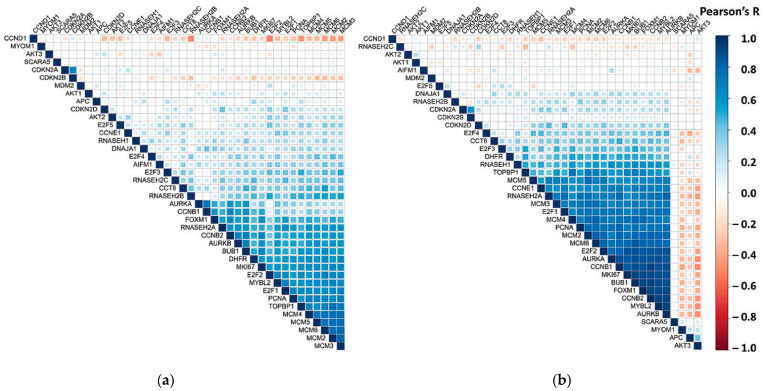
*RNAESH2A* correlation with 39 other genes previously listed in CCLE and TCGA Pan Cancer datasets: (**a**) correlation coefficient plot generated from RNA-Seq expression levels in 1019 cancer cell lines from CCLE database measured in RPKM (log2 (value + 1)); (**b**) correlation coefficient plot generated from RNA-Seq expression levels in 10,071 cancer tissue samples from the Cancer Genome Atlas (TCGA) Pan Cancer studies measured in RSEM (Bach normalized from Illumina HiSeq_RNASeqV2(log2 (value + 1)). The proportion of shaded area in each cell corresponds to the Pearson’s R.

**Table 1 biology-10-00221-t001:** Gene ontology (GO) term analysis of the top 2% co-expressed genes of *RNASEH2A* reveals the possible role of *RNASEH2A* in cell cycle regulation. The analysis shows *RNASEH2A* functional networks based on process (blue panel), function (orange panel), and component (green panel). The analysis was made by Gene Ontology enRichment anaLysis and visuaLizAtion tool (GOrilla). Statistic is shown by both *p*-value and false discovery rate (FDR).

**Process**			
**GO Term**	**Description**	***p*-Value**	**FDR *q*-Value**
GO 1903047	mitotic cell cycle process	2.80 × 10^−14^	5.93 × 10^−11^
GO 0070507	regulation of microtubule cytoskeleton organ	1.39 × 10^−7^	8.40 × 10^−5^
GO 0006974	cellular response to DNA damage stimulus	2.77 × 10^−5^	5.86 × 10^−3^
GO 0007059	chromosome segregation	9.00 × 10^−5^	1.36 × 10^−2^
GO 0006260	DNA replication	2.66 × 10^−4^	3.12 × 10^−2^
**Function**			
**GO Term**	**Description**	***p*-Value**	**FDR *q*-Value**
GO 0140097	catalytic activity, acting on DNA	9.59 × 10^−6^	9.06 × 10^−3^
GO 0008017	microtubule binding	4.52 × 10^−5^	1.42 × 10^−2^
GO 0019900	kinase binding	3.28 × 10^−4^	7.74 × 10^−2^
**Component**			
**GO Term**	**Description**	***p*-Value**	**FDR *q*-Value**
GO 0044427	chromosomal part	4.06 × 10^−8^	1.36 × 10^−5^
GO 0000922	spindle pole	1.13 × 10^−6^	2.52 × 10^−4^
GO 0005654	nucleoplasm	3.52 × 10^−6^	5.88 × 10^−4^
GO 0030496	midbody	5.01 × 10^−6^	6.70 × 10^−4^
GO 0005815	microtubule organizing center	5.13 × 10^−6^	5.72 × 10^−4^
GO 0000940	condensed chromosome outer kinetochore	8.72 × 10^−6^	7.28 × 10^−4^
GO 0000796	condensin complex	8.71 × 10^−5^	4.16 × 10^−3^

**Table 2 biology-10-00221-t002:** *RNASEH2A* is highly expressed in human proliferative tissue. Expression of *RNASEH* genes in different 53 human tissues. Data obtained from the Genotype-Tissue Expression portal (GTEx) consist of the median transcripts per million (TPM) from >90 individuals for each tissue. Blue color indicates low expression levels and red color indicates high expression levels.

	*RNASEH1*	*RNASEH2A*	*RNASEH2B*	*RNASEH2C*		*RNASEH1*	*RNASEH2A*	*RNASEH2B*	*RNASEH2C*
Liver	5.193	3.947	3.878	11.57	Adipose-Subcutaneous	13.845	15.96	16.585	30.145
Muscle-Skeletal	11.72	5.9535	5.51	9.215	Lung	12.92	14.51	19.34	30.99
Heart-Left Ventricle	7.251	4.262	8.466	12.71	Breast-Mammary Tissue	12.93	16.39	16.65	33.895
Pancreas	5.3175	6.44	10.175	11.48	Pituitary	8.271	11.02	14.5	52.95
Whole Blood	2.726	7.958	6.471	16.33	Vagina	12.08	20.33	16.85	39.11
Brain-Putamen (basal ganglia)	4.6985	5.5965	7.875	15.635	Esophagus-Muscularis	14.91	8.687	22.84	46.885
Brain-Caudate (basal ganglia)	5.1025	6.1935	9.2325	17.175	Artery-Tibial	16.91	8.638	17.8	50.75
Brain-Substantia nigra	6.0725	7.4295	8.123	16.355	Thyroid	12.615	16.935	24.905	43.205
Brain-Amygdala	5.0275	7.4755	8.2985	17.285	Esophagus-Gastroesophageal Junction	13.75	8.3635	22.05	53.92
Brain-Hippocampus	5.907	6.655	7.567	18.76	Bladder	15.4	15.95	23.68	43.32
Brain-Anterior cingulate cortex (BA24)	5.921	6.326	9.554	19.13	Artery-Coronary	13.65	10.5	18.58	56.4
Brain-Hypothalamus	6.903	7.451	8.639	18.47	Artery-Aorta	14.63	9.235	17.31	58.05
Kidney-Cortex	6.303	6.632	6.614	22.18	Prostate	11.11	11.475	20.27	57.625
Brain-Nucleus accumbens (basal ganglia)	6.557	7.162	12.03	20.17	Nerve-Tibial	15.18	19.24	19.55	48.22
Heart-Atrial Appendage	8.609	5.969	9.644	22.33	Fallopian Tube	14.57	13.31	25.31	49.21
Brain-Cortex	5.5355	6.8145	11.095	25.555	Colon-Sigmoid	13.24	10	24.46	55.42
Brain-Frontal Cortex (BA9)	7.597	6.766	13.7	23.46	Cells-Transformed fibroblasts	29.98	34.04	17.95	25.17
Stomach	8.19	9.396	12.55	24.365	Brain-Cerebellum	9.384	16.21	12.41	70.97
Brain-Spinal cord (cervical c-1)	9.183	13.54	10.54	22.75	Spleen	10.935	25.695	24.555	49.11
Skin-Sun Exposed (Lower leg)	11.62	18.75	10.87	19.18	Ovary	12.4	19.46	20.07	59.51
Skin-Not Sun Exposed (Suprapubic)	10.72	18.3	11.7	19.98	Cervix-Ectocervix	14.765	24.625	23.54	51.435
Minor Salivary Gland	10.33	13.21	13.03	24.15	Brain-Cerebellar Hemisphere	13.325	17.52	14.98	68.75
Esophagus-Mucosa	10.58	24.62	11.92	18.2	Cervix-Endocervix	15.15	19	26.62	55
Colon-Transverse	9.484	12.52	14.885	28.96	Testis	14.12	49.84	23.42	31.66
Adrenal Gland	10.045	9.7865	12.39	35.965	Uterus	16.6	21.33	30.43	66.03
Adipose-Visceral (Omentum)	12.25	12.14	14.71	32.23	Cells-Epstein-Barr virus (EBV)-transformed lymphocytes	27.94	79.945	41.36	32.195
Small Intestine-Terminal Ileum	9.633	14.12	18.32	30.46					

**Table 3 biology-10-00221-t003:** Percentage of patients with *RNASEH2A* copy number alterations from Genomic Identification of Significant Targets in Cancer (GISTIC) in The Cancer Genome Atlas (TCGA) Pan Cancer cohorts.

Cancer Type	Percentage of Patients with Copy Number Alterations				
	Deep Deletion	Shallow Deletion	Diploid	Gain	Amplification
Ovarian Epithelial Tumor	0.34%	29.15%	26.10%	36.27%	8.14%
Endometrial Carcinoma	-	12.13%	69.67%	14.90%	3.29%
Adrenocortical Carcinoma	-	1.32%	34.21%	61.84%	2.63%
Pleural Mesothelioma	-	8.05%	73.56%	16.09%	2.30%
Esophageal Squamous Cell Carcinoma	-	34.04%	44.68%	19.15%	2.13%
Cervical Squamous Cell Carcinoma	0.41%	26.23%	57.79%	13.52%	2.05%
Diffuse Glioma	0.20%	3.33%	73.92%	20.78%	1.76%
Sarcoma	-	10.36%	50.20%	37.85%	1.59%
Invasive Breast Carcinoma	0.09%	21.25%	60.11%	17.23%	1.31%
Ocular Melanoma	-	3.75%	92.50%	2.50%	1.25%
Thymic Epithelial Tumor	-	2.52%	94.12%	2.52%	0.84%
Esophagogastric Adenocarcinoma	0.62%	33.40%	57.73%	7.42%	0.82%
Head and Neck Squamous Cell Carcinoma	0.20%	19.45%	66.99%	12.57%	0.79%
Glioblastoma	-	8.11%	58.11%	33.11%	0.68%
Non-Small Cell Lung Cancer	0.30%	44.10%	43.39%	11.71%	0.50%
Bladder Urothelial Carcinoma	-	29.21%	53.71%	16.58%	0.50%
Prostate Adenocarcinoma	0.20%	5.74%	92.21%	1.64%	0.20%
Cervical Adenocarcinoma	2.17%	26.09%	65.22%	6.52%	-
Cholangiocarcinoma	-	13.89%	72.22%	13.89%	-
Colorectal Adenocarcinoma	0.51%	11.02%	71.86%	16.61%	-
Encapsulated Glioma	-	-	100.00%	-	-
Fibrolamellar Carcinoma	-	-	100.00%	-	-
Hepatocellular Carcinoma	0.28%	21.51%	64.25%	13.97%	-
Leukemia	-	1.81%	95.18%	3.01%	-
Mature B-Cell Neoplasms	-	2.08%	91.67%	6.25%	-
Melanoma	-	22.62%	61.31%	16.08%	-
Miscellaneous Neuroepithelial Tumor	-	-	96.30%	3.70%	-
Non-Seminomatous Germ Cell Tumor	-	52.33%	33.72%	13.95%	-
Pancreatic Adenocarcinoma	-	14.77%	77.27%	7.95%	-
Pheochromocytoma	-	0.75%	84.33%	14.93%	-
Renal Clear Cell Carcinoma	-	2.17%	88.56%	9.27%	-
Renal Non-Clear Cell Carcinoma	-	5.17%	87.36%	7.47%	-
Seminoma	-	1.59%	55.56%	42.86%	-
Undifferentiated Stomach Adenocarcinoma	-	25.00%	58.33%	16.67%	-
Well-Differentiated Thyroid Cancer	-	1.21%	97.37%	1.41%	-
All Cancer types	0.17%	17.69%	66.83%	14.30%	1.01%

CNAs are categorized as: 1. Deep deletion (−2) indicating a deep loss/homozygous deletion; 2. Shallow deletion (−1) indicates shallow loss/heterozygous deletion; 3. Diploid (0) indicates homozygous genes; 4. Gain indicates a low-level gain (a few additional copies, often broad); 5. Amplification (2) indicate a high level amplification (more copies, often focal).

**Table 4 biology-10-00221-t004:** List of proteins identified as binding partners of *RNASEH2A* by mass spectrometry analysis. The two top proteins, RNASEH2B and RNASEH2C, confirm the authenticity of the results.

Protein	Gene	Description
Q5TBB1	RNASEH2B	Ribonuclease H2 subunit B
E9PN81	RNASEH2C	Ribonuclease H2 subunit C
P31689	DNAJA1	DnaJ homolog subfamily A member 1
O95831	AIFM1	Apoptosis-inducing factor 1, mitochondrial
P50990	CCT8	T-complex protein 1 subunit theta

**Table 5 biology-10-00221-t005:** List of genes related to cancer proliferation and cell cycle phases used for correlation analysis in Cancer Cell Line Encyclopedia (CCLE) and TCGA Pan Cancer datasets.

Gene Name (s)	Function/Role	References
*CDKN2A*, *CDKN2B*, *CCND1*	G1 Cell Cycle Phase	A. Subramanian et al. Gene Set Enrichment Analysis (GSEA Database) [46]
*DHFR*, *CCNE1*	G1/S Cell Cycle Phase	A. Subramanian et al. (GSEA Database) [46]
*AKT1-3*, *E2F4-5*	S Cell Cycle Phase	A. Subramanian et al. (GSEA Database) [46]
*CDKN2D*, *MDM2*	G2 Cell Cycle Phase	A. Subramanian et al. (GSEA Database) [46]
*CCNB2*, *TOPBP1*	G2/M Cell Cycle Phase	A. Subramanian et al. (GSEA Database) [46]
*APC*, *BUB1*	M Cell Cycle Phase	A. Subramanian et al. (GSEA Database) [46]
*E2F2*, *E2F3*, *CCNB1*	M/G1 Cell Cycle Phase	A. Subramanian et al. (GSEA Database) [46]
*MYBL2*, *FOXM1*, *BUB1*, *AURKA*, *AURKB*	Upregulated in cancer	M. Li et al. [40]
*SCARA5*, *MYOM1*	Downregulated in cancer	M. Li et al. [40]
*PCNA, MKI67(Ki67)*, *MCM2–MCM6*, *E2F1*	Proliferative markers in cancer	M.L. Whitfield et al. [47]
*CCNE1*, *CCND1*, *CCNB1*	Cell cycle markers associated with cancer (G1/S, G2, and M)	M.L. Whitfield et al. [47]
*RNASEH2A*, *RNASEH2B*, *RNASEH2C, RNASEH1*	Target genes in this study	This study
*CCT8*, *DNAJA1*, *AIFM1*	Predicted Binding partners of *RNASEH2A*	This study

## Data Availability

The raw data of the study are presented within the Appendix A and are also available from the corresponding author.

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
