# Peer review of "Gene Co-Expression Analysis of Human RNASEH2A Reveals Functional Networks Associated with DNA Replication, DNA Damage Response, and Cell Cycle Regulation"

_biology, 2021, doi:10.3390/biology10030221_

Round 1

Reviewer 1 Report

The manuscript is satisfactorily revised. there are no further comments however spell check/sentence structure should be checked.   

Reviewer 2 Report

none.

Reviewer 3 Report

The authors have addressed my concerns, and I believe that the manuscript is ready for publication.

This manuscript is a resubmission of an earlier submission. The following is a list of the peer review reports and author responses from that submission.

Round 1

Reviewer 1 Report

This is a well written manuscript with clear presentation of the data. Authors have performed in-silico analysis to investigate the role of RNASEH2A in various cellular processes. Authors have made attempts to validate their in-silico findings using HEK293 cells, in-vitro.

There are several points that need to be addressed before the publication of this manuscript as noted below:

1. Did the authors look at the copy number alterations for RNASEH2A in various tissues and cancers? 

2. Did the authors query the CCLE data set for RNASEH2A gene? This could be a good source to investigate the relationship of RNASEH2A with growth/proliferation rate and coexpression levels at cell line level. This could have also guided the authors to select an appropriate cell line for their in-vitro validation studies. 

3. When selecting the patient data for analysis did the authors consider factors like therapy/treatment, age, surgery etc? And would these factors have had any effect on the outcome of the analysis presented in this manuscript? 

4. Why did the authors choose to use HEK293 cells and not any of the lung or breast cancer cell lines or a pair of normal and cancer cell lines? which could have been more appropriate for this manuscript. Or perhaps choose 2-3 cell lines to validate the in-silico findings? Using HEK293 cells, and only one cell line, could not be considered a strong evidence of validation. Authors are not requested to repeat the experiments again in other cancer cell lines, which is actually more appropriate, but authors must clarify the choice of their cell line and how the data obtained from HEK293 could be applied overall to other cell/cancer types. It also recommended that Author note the limitations of their data as suggested in the points below. 

5. Under discussion, authors highlight that cell cycle is an example, does the in-vitro data from HEK293 cells provides any correlation to the information obtained from in-silico analysis? How the data from HEK293 has been useful to validate the in-silico findings? 

6. it is recommended that authors must include, under discussion section, the limitations/short comings of their analysis/study. And also include how the findings in this manuscript could be applied in further research. 

Author Response

Reviewer 1.

  1. Did the authors look at the copy number alterations for RNASEH2A in various tissues and cancers?

We looked at copy number alterations (CNA) in RNASEH2A in TCGA dataset, which calculated the copy number variation using GISTIC algorithm where the CNA categories defined are as deep deletion(-2), shallow deletion(-1), diploid(0), gain(+1) and amplification(+2) in the cancer patients and reported percentage of patients having these copy number variations in each cancer type/subtype. Our findings suggest that a subset of cancer types/subtypes show high amplification (>1%) as compared to deep deletion, suggesting higher RNASEH2A expression is associated with cancer prevalence for those subset cancer types. We also find Breast invasive carcinoma amongst the list of cancer types having higher prevalence of amplifications. The results are represented in Table 3 of the revised manuscript.

  1. Did the authors query the CCLE data set for RNASEH2A gene? This could be a good source to investigate the relationship of RNASEH2A with growth/proliferation rate and co-expression levels at cell line level. This could have also guided the authors to select an appropriate cell line for their in-vitro validation studies.

We thank the reviewer for their suggestion which lead to further validation of the findings of our study. As our original correlation was defined in tissues, we looked at RNA expression in both CCLE dataset and TCGA Pan Cancer Studies. Here we picked RNASEH2A + 39 other genes involved in cancer progression/proliferation, cell cycle associated genes for validating GO analysis, and proteins identified by mass spectrometry analysis in this study. We found that RNASEH2A has higher correlations with cancer proliferation and certain cell cycle genes as compared to RNASEH2B, RNASEH2C and RNASEH1 suggesting that RNASEH2A might have an important role to play as compared to other subunits of the RNASEH2 complex in cancer biology. The results are represented by Figure 5 and raw data is in S15 a-b.

  1. When selecting the patient data for analysis did the authors consider factors like therapy/treatment, age, surgery etc? And would these factors have had any effect on the outcome of the analysis presented in this manuscript?

Cancer RNA-seq Nexus database provides information exclusively on differential expression in genes amongst cancer vs normal, including different stages of cancer for few cancer types. This is one of the limitations of our study as we use a dataset where patient information is not provided.

  1. Why did the authors choose to use HEK293 cells and not any of the lung or breast cancer cell lines or a pair of normal and cancer cell lines? which could have been more appropriate for this manuscript. Or perhaps choose 2-3 cell lines to validate the in-silico findings? Using HEK293 cells, and only one cell line, could not be considered a strong evidence of validation. Authors are not requested to repeat the experiments again in other cancer cell lines, which is actually more appropriate, but authors must clarify the choice of their cell line and how the data obtained from HEK293 could be applied overall to other cell/cancer types. It also recommended that Author note the limitations of their data as suggested in the points below.

We chose to work with the HEK293 cell line as we wanted to use a system easy to transfect in order to get a reasonable amount of the RNASEH2A protein for immunoprecipitation experiments and mass spectrometry analysis. There is a debate in the field regarding on how this cell line should be considered, normal versus cancer cells. As we used a high passage of this cell line, and because it has been shown that high passage HEK293 cells show a cancer phenotype, we are confident to have worked with a cell line similar to a cancer cell line. In addition, it has been shown that over-expression experiments change the phenotype of this cell line from normal to a cancer type. As we have shown that the proteins identified by mass spectrometry have a high correlation with RNASEH2A and with genes upregulated in cancers, we are confident that our results can be reproducible using other cancer cell lines/tissues, where the protein level of RNASEH2A is high. These cell/tissue systems will definitely help addressing the biology role of RNASEH2A in cancer by identifying new signaling pathways in which RNASEH2A is involved. We have included the discussion related to use of HEK293 in lines 566-568.

  1. Under discussion, authors highlight that cell cycle is an example, does the in-vitro data from HEK293 cells provides any correlation to the information obtained from in-silico analysis? How the data from HEK293 has been useful to validate the in-silico findings?

To further explore the involvement of RNASEH2A in cancer, we examined the correlations between RNASEH2A and 39 additional genes in CCLE and TGCA Pan Cancer datasets (Table 5 and Table S14 a-b). The correlation analysis uncovered a positive correlation of RNASEH2A with genes up-regulated in cancer and a negative correlation with genes that are instead down-regulated. In addition, RNASEH2A showed a higher correlation with cluster of genes known to play a role in cancer proliferation as compared to RNASEH2B, RNASE2C and RNASEH1. We also observed a similar trend for the three putative binding partners of RNASEH2A that we identified by mass spectrometry, showing a correlation comparable with that found in the GTEx dataset. The analysis of cell cycle related genes showed a high correlation between RNASEH2A and genes involved in G1/S, G2, G2/M, M and M/G1 phases; and a comparatively lower or no correlation with genes unique to G1 and S phases (Figure 5 and raw data in S15 a-b).

  1. it is recommended that authors must include, under discussion section, the limitations/short comings of their analysis/study. And also include how the findings in this manuscript could be applied in further research.

We have added the limitations/short comings of our analysis at the end of the Discussion, as suggested by the reviewer (lines 560-571). We addressed how the findings in this manuscript could be applied in further research in the conclusion (lines 576-578).

Reviewer 2 Report

In the manuscript (MS#1090564), Marsili et al. performed co-expression analysis of RNASEH2A, the catalytic subunit of Ribonuclease (Rnase) H2, using bioinformatics approach combined with some sort of biochemical approach. Although RNASEH2 is critical for the removal of RNA in RNA-DNA hybrids, the exact function and mechanism of RNASEH2A in genome integrity and cancer biology/etiology remains unknown.

Their GO term analysis reveals that the top 2% co-expressed genes of RNASEH2A is associated with cell cycle regulation (process, function and component). It is further demonstrated that the RNASEH2 expression is found up-regulated in actively cycling cells and human tissues. Interestingly, RNASEH2A is the only gene upregulated in 29 cancer tissues out of all RNASEH genes (1, 2A, 2B, 2C, and 3). Next, the authors utilized mass spectrometry analysis with GFP-RNASEH2A and identified a list of RNASEH2A-interacting proteins such as CCT8, DNAJA1, and AIFM1 in addition to expected RNASEH2B and RNASEH2C.

This study is designed well and with good rationales. The writing of the manuscript is good. The quality and significance of this study can be improved by addressing below major and minor issues before recommendation of publication for Biology.

Major issues:

  • In the first paragraph on Page 8, it is stated that “In all these cancers RNASEH2A was upregulated at early stage and remained high with the progression of the disease (Figure 3 and Table S10 and raw data in Table S11). No other gene, beside MYBL2, was upregulated at the different stages studied. These results imply that RNASEH2A might have a role in cell proliferation and cancer progression that is dependent on its expression level.” This observation may be interpreted differently that its current version. It seems to this reviewer that RNASEH2A upregulation may play a role in the cancer etiology or transformation from normal tissue to cancerous tissue, rather than cancer progression. Thus, more clarification or better description/interpretation is needed.

  • It is of significant to show some biochemical evidence that RNASEH2A interacts with newly identified interacting proteins (CCT8, DNAJA1, and AIFM1), although the correlation analysis of these three genes with RNASEH2A is high. Such biochemical experiments will validate the new protein-protein interaction of RNASEH2A with its binding partners. At least one GFP immunoprecipitation experiment showing GFP-RNASEH2A but not GFP interacts with its new binding partner in place will be helpful for the validation of the mass spec results.

Minor issues:

  • The title may be edited a little bit to be consistent with the Journal requirement and format such as “Gene Co-Expression Analysis of Human RNASEH2A Reveals Functional Networks Associated with DNA Replication, DNA Damage Response, and Cell Cycle Regulation”
  • In Figure 3, it seems that some portion of its right side is cut somehow. It may need to reinsert Figure 3 to the word template file.
  • In Figure 4, an overall summary of the whole figure is missing. It is highly recommended to add lane numbers in all panels in (a) and (b) and to cite such lanes in the text and figure legend.

Author Response

Reviewer 2.

Major issues:

  • In the first paragraph on Page 8, it is stated that “In all these cancers RNASEH2A was upregulated at early stage and remained high with the progression of the disease (Figure 3 and Table S10 and raw data in Table S11). No other gene, beside MYBL2, was upregulated at the different stages studied. These results imply that RNASEH2A might have a role in cell proliferation and cancer progression that is dependent on its expression level.” This observation may be interpreted differently that its current version. It seems to this reviewer that RNASEH2A upregulation may play a role in the cancer etiology or transformation from normal tissue to cancerous tissue, rather than cancer progression. Thus, more clarification or better description/interpretation is needed.

We agree with the reviewer comments. Indeed, our findings suggest that RNASEH2A up-regulation is a key player in cancer transformation. We have clarified and better described the first paragraph of Page 8.

  • It is of significant to show some biochemical evidence that RNASEH2A interacts with newly identified interacting proteins (CCT8, DNAJA1, and AIFM1), although the correlation analysis of these three genes with RNASEH2A is high. Such biochemical experiments will validate the new protein-protein interaction of RNASEH2A with its binding partners. At least one GFP immunoprecipitation experiment showing GFP-RNASEH2A but not GFP interacts with its new binding partner in place will be helpful for the validation of the mass spec results.

We understand that additional biochemical experiments such as reciprocal immunoprecipitations would further validate the interactions between RNASEH2A and the putative binding partners (CCT8, DNAJA1, and AIFM1) identified in our study. As discussed above, in response to point 2 or Reviewer #1, to further explore the involvement of RNASEH2A in cancer, we examined the correlations between RNASEH2A and 39 additional genes in CCLE and TGCA Pan Cancer datasets (Table 5 and Table S14 a-b). The correlation analysis uncovered a positive correlation of RNASEH2A with genes up-regulated in cancer and a negative correlation with genes that are instead down-regulated. In addition, RNASEH2A showed a higher correlation with cluster of genes known to play a role in cancer proliferation as compared to RNASEH2B, RNASE2C and RNASEH1. We also observed a similar trend for the three putative binding partners of RNASEH2A that we identified by mass spectrometry, showing a correlation comparable with that found in the GTEx dataset. We believe that such analyses using the CCLE and TGCA Pan Cancer datasets provide support for our findings.

Minor issues:

  • The title may be edited a little bit to be consistent with the Journal requirement and format such as “Gene Co-Expression Analysis of Human RNASEH2A Reveals Functional Networks Associated with DNA Replication, DNA Damage Response, and Cell Cycle Regulation”

We have made the requested correction and now the title is consistent with the Journal requirements and format.

  • In Figure 3, it seems that some portion of its right side is cut somehow. It may need to reinsert Figure 3 to the word template file.

We submitted our manuscript on January 11, 2021, using a 2020 Biology template which it was then converted to a 2021 Biology template by the Editorial office. This explains why the right part of Figure 3 was cut somehow. We have now addressed this issue and Figure 3 is now visible in each of its part.

  • In Figure 4, an overall summary of the whole figure is missing. It is highly recommended to add lane numbers in all panels in (a) and (b) and to cite such lanes in the text and figure legend.

We have added an overall summary to Figure 4 (line 415). We also added lane numbers in all panels in (a) and (b) at the bottom of each gel image. Lane numbers has been cited in text (lines 398, 400-401) and figure legend (lines 417-418, 421-422).

Reviewer 3 Report

This is an interesting study and the authors have collected a dataset using the co-expression correlation approach. The paper is generally well written and structured. However, there are a number of minor comments that the authors should consider.

  1. Title: DNA instead of Dna

  1. In the section of protein analysis (Western blot). Line 152 specific primary antibodies instead of antibody. Line 154 “After washing 3x with TBS-T”. It is better to mention the time, for example, “after washing 3x 5 or 10 mins with TBS-T”. It also could be improved by deleting the (#cat and #lot) for each item.

  1. In figure 3 the graphs and data have overlapped the page and some data appears to be missing because of this.
  2. There are a few typographical errors throughout the manuscript. For example, line 11 space between H2 ternary and complex. Line 35, “with 30 µl” instead of “with 30 ul”, and others.

  1. The authors stated that “Tissues part of the reproduction system had a higher expression levels of RNASEH genes, while low levels were found mainly in tissues with low proliferation and cell turnover capacity”. Any more information could be added to this significant statement and explain what they mean for future research?

Author Response

Reviewer 3.

Minor comments:

  1. Title: DNA instead of Dna

We have addressed this issue in the title, and Dna has been changed to DNA.

  1. In the section of protein analysis (Western blot). Line 152 specific primary antibodies instead of antibody. Line 154 “After washing 3x with TBS-T”. It is better to mention the time, for example, “after washing 3x 5 or 10 mins with TBS-T”. It also could be improved by deleting the (#cat and #lot) for each item.

In the section of protein analysis (western Blot), we changed “antibody” to antibodies (line 163). We changed “After washing 3x with TBS-T” to “after washing 3x 10 minutes with TBS-T” (lines 164,167). To improve this section, we have deleted the #cat and #lot for each item.

  1. In figure 3 the graphs and data have overlapped the page and some data appears to be missing because of this.

We submitted our manuscript on January 11, 2021, using a 2020 Biology template which it was then converted to a 2021 Biology template by the Editorial Office. This explains why the right part of Figure 3 was cut somehow. We have now addressed this issue, and Figure 3 is now visible in each of its part.

  1. There are a few typographical errors throughout the manuscript. For example, line 11 space between H2 ternary and complex. Line 35, “with 30 μl” instead of “with 30 ul”, and others.

We have made the corrections and removed line space between H2 ternary and complex (line 11) and changed “with 30ul” to “with 30μl”, (line 146). A complete spelling and grammar check was done for the entire manuscript.

  1. The authors stated that “Tissues part of the reproduction system had a higher expression levels of RNASEH genes, while low levels were found mainly in tissues with low proliferation and cell turnover capacity”. Any more information could be added to this significant statement and explain what they mean for future research?

The reproductive system undergoes a rapid cellular turnover throughout adult life and this biologic event has been linked to the existence of adult stem cells in the genital tract responsible of tissue regeneration. Our results suggests that RNASEH2A expression is high in these tissues. Therefore, the existence of adult stem cells in these tissues is probably responsible for the high expression of RNASEH2A. Indeed, deregulation of the adult stem cell niche is considered a potentially significant step in the etiology of cancer as well as other proliferative disorders. Our results support the involvement of RNASEH2A in the etiology of cancer as we showed that copy number alterations such as RNASEH2A amplifications occur in a maximum percentage of patients having ovarian epithelium tumor and endometrial carcinoma. Therefore, basic research in the biology of adult stem cells and the characterization of RNASEH2A function and de-regulation in these cells would be a promising field in the search of novel therapeutic targets for these types of cancer. We have modified the text on page 17, from line 535 to 539 to include such discussion of our results: << Interestingly, cancers from tissues with high cell turnover, such as the reproductive system, showed a maximum percentage of patients with RNASEH2A amplification, suggesting a possible role of adult stem cells in the overexpression of RNASEH2A. In fact, deregulation of the adult stem cell niche is considered a key event in the etiology of cancer >>.